# Compositional Changes in the Vaginal Bacterial Microbiome of Healthy Pregnant Women across the Three Gestational Trimesters in Ismailia, Egypt

**DOI:** 10.3390/microorganisms11010139

**Published:** 2023-01-05

**Authors:** Mariam E. Amin, Marwa Azab, Amro Hanora, Khaled Atwa, Sarah Shabayek

**Affiliations:** 1Department of Microbiology and Immunology, Faculty of Pharmacy, Suez Canal University, Ismailia 41522, Egypt; 2Department of Obstetrics and Gynecology, Faculty of Medicine, Suez Canal University, Ismailia 41522, Egypt

**Keywords:** vaginal microbiome, pregnancy trimesters, *Firmicutes*, *Lactobacillus iners*, V3–V4, Ismailia, Egypt, healthy

## Abstract

The composition of the vaginal microbiome may lead to adverse pregnancy outcomes. Normal pregnancy is associated with changes in the vaginal bacterial community composition, which tend to be more enriched with one or two *Lactobacillus* species promoting a healthy vagina and favorable birth outcomes. The aim of the current study was to determine compositional changes in the healthy vaginal microbiome composition during the three trimesters of pregnancy in Ismailia, Egypt using Illumina MiSeq sequencing of the V3–V4 region of the 16S rRNA. The phylum *Firmicutes* and the genus *Lactobacillus* dominated across the three trimesters of pregnancy. *L. iners* was the most abundant species. However, *L. coleohominis* and *L. reuteri* represented the least dominant vaginal lactobacilli. Core microbiome analyses showed the *Lactobacillus* genus and *L. iners* species to have the highest prevalence in all the samples of our study groups. The phylum *Firmicutes* was found to be negatively correlated with almost all other vaginal phyla during pregnancy. Likewise, a negative correlation between *Lactobacillus* and almost all other genera was detected, including significant negative correlations with *Dialister* and *Prevotella.* Furthermore, negative correlations of *L. iners* were detected with almost all other species, including a significant negative correlation with *L. helveticus, G. vaginalis, S. anginosus*, and *S. agalactiae.*

## 1. Introduction

Recently, the number of studies investigating the composition of the vaginal ecosystem and their impact on female reproductive health is increasing [1,2,3,4,5,6,7,8,9,10,11,12,13,14,15,16]. Next-generation sequencing technology provides a detailed profiling of the human vaginal microbiome; however, the exact mechanisms defining its role in women’s reproductive health remains unknown. Vaginal microbiome is crucial for both maternal and neonatal health during pregnancy [17]. 

The vaginal microbiome of healthy pregnant women tends to be more enriched and dominated with one or two *Lactobacillus* species compared to the non-pregnant state [18,19]. The hormone levels in pregnancy favor the dominance of *Lactobacillus* species while *Gardnerella vaginalis* and other anaerobic bacteria decrease. Pregnancy is associated with high levels of circulating estrogen produced from the ovaries and the placenta [5,20]. The elevated estrogen levels promote glycogen deposition in the vaginal epithelial cells, which, in turn, promotes the proliferation of different *Lactobacillus* species. Due to these changes, the composition of the vaginal microbiome is stable with limited variations during pregnancy [5,17,21,22]. Such dominance of *Lactobacillus* species has been linked to the low production of proinflammatory cytokine, healthy vagina, and favorable birth outcomes [5,23,24].

Changes in the vaginal microbiome composition during pregnancy commonly include transitions between different *Lactobacillus* species [3,18]. Five community state types (CSTs) of bacteria have been described in asymptomatic women at a reproductive age between 12 and 45 years [5,15,16]. These are CST I, which is dominated by *L. crispatus*; CST II, which is dominated by *L. gasseri*; CST III, which is dominated by *L. iners*; and CST V, which is dominated by *L. jensenii*, while CST IV is depleted in *Lactobacillus* species and enriched with diverse anaerobic bacteria, such as *Atopobium vaginae*, *G. vaginalis, Prevotella, Megasphaera, Dialister, Sneathia, Peptoniphilus, Mobiluncus*, and *Finegoldia*. These are usually reported in association with bacterial vaginosis (BV) and increased risk of preterm birth [25,26].

The composition of the vaginal microbiome may also be shaped through ethnic, genetic, cultural, and behavioral differences between hosts [10,14,15,27]. Vaginal microbiomes dominated by *L. iners* or with more diverse anaerobic communities, mostly *G. vaginalis* and *Candidatus Lachnocurva vaginae*, which is formerly known as bacterial vaginosis-associated bacterium 1 (BVAB1), have been reported for most sub-Saharan African pregnant women [5,15,27]. However, studies describing vaginal microbiomes in Egypt or nearby regions are still sparse. The aim of the present study was to investigate the vaginal microbiome composition of healthy pregnant women in Egypt during the first, second, and third trimesters compared to healthy non-pregnant women. The potential correlations between the vaginal microbiota during pregnancy versus non-pregnancy state have also been demonstrated. This has been achieved through next-generation sequencing of the V3–V4 hypervariable region of the 16S rRNA gene. 

## 2. Materials and Methods

### 2.1. Ethical Statement

This study was approved by the scientific research ethics committee of the Faculty of Pharmacy, Suez Canal University, Ismailia, Egypt (Reference number 201803PHDH1). The study was conducted in accordance with the principles of Helsinki Declaration. Informed consent was obtained from all women prior to enrollment.

### 2.2. Participant Enrollment

We analyzed samples from 36 pregnant women and 12 non-pregnant women attending the obstetrics and gynecology clinics in Suez Canal University Educational Hospital and El Sheikh Zayed Governmental Health Office from July 2020 to October 2020. All women were healthy with no known medical issues. Participants were sampled cross sectionally and were divided into four study groups: (1) healthy non-pregnant women (control group) (*n* = 12), (2) healthy pregnant women in the first trimester of pregnancy (*n* = 12), (3) healthy pregnant women in the second trimester of pregnancy (*n* = 12), (4) healthy pregnant women in the third trimester of pregnancy (*n* = 12). Inclusion and exclusion criteria were employed, as has been previously described [10,28,29,30]. All participants were aged between 18–40 years (Appendix A) with regular menstruation and had no sexual intercourse, tampon usage, douching, or vaginal creams in the preceding 72 h. All non-pregnant participants had suffered no bacterial vaginosis and had a vaginal pH ≤ 4.5. All pregnant women showed normal singleton gestation with no maternal comorbidities, such as gestational diabetes type A2, diabetes mellites, hypertension, or any other chronic diseases. Participants were excluded for the following reasons: smoking, alcohol intake, bacterial vaginosis, principle dietary changes, history of cancer, immunocompromised status, endocrine disturbances, medication usage within the last 6 months, e.g., antibiotics, hormones, corticosteroids, cytokines, probiotics. The participants were divided according to pregnancy status into four groups: non-pregnant, pregnant during the first trimester, pregnant during the second trimester, and pregnant during the third trimester. Obstetrical dating for gestational age was determined, as indicated by the American College for Obstetricians and Gynecologists (ACOG) (https://www.acog.org/, accessed on 23 December 2022). First trimester is from first day of last menstrual period (LMP) to 13 weeks and 6 days. Second trimester is from 14 weeks and 0 days to 27 weeks and 6 days. Third trimester is from 28 weeks and 0 days to 40 weeks and 6 days.

### 2.3. Sample Collection

Vaginal samples were obtained using sterile cotton swabs inserted about 4 to 5 cm into the vagina, twisted and wiped in full circles in order to collect vaginal material on all sides of the tip, and kept in the vagina for 20 s [31,32,33]. Afterwards, the swab was carefully removed and placed in a sterile Eppendorf tube containing 1 mL of sterile phosphate-buffered saline (PBS) (Sigma, Darmstadt, Germany, Cat. No. 806552, pH 7.4) with vigorous shaking [29,34]. Samples were transported on ice and immediately stored at −80 °C [9,29,34,35,36].

### 2.4. The pH Measurement

Vaginal pH of non-pregnant women (control group) was measured using commercial pH strips in the pH range 4.5–10.0 (Sigma, Germany, Cat. No. P4536). The color of the test pH indicator was compared to a standard color chart provided with the pH applicator. Non-pregnant women with pH exceeding 4.5 were excluded from the study [21,28,37]. As recommended by Agaard et al., 2012 [28], vaginal pH was not recorded for pregnant participants; however, they were excluded if presenting any signs and symptoms for bacterial vaginosis.

### 2.5. DNA Extraction 

Samples were released from −80 °C conditions and allowed to reach room temperature. Microbial DNA was extracted using Dneasy PowerSoil DNA extraction kit (Qiagen, Valencia, CA, USA, Cat. No. 12888-50) [38,39], according to manufacturer’s instructions. Extraction steps were conducted after gentle vortex of the collection tubes containing swab tips to induce better dispersion of the trapped bacteria.

### 2.6. PCR Amplification of 16S rRNA Gene and MiSeq Illumina Sequencing

PCR targeting amplification of the hypervariable regions V3–V4 of 16S rRNA gene was performed according to the Illumina 16S rDNA Metagenomic Sequencing Library Preparation protocol (Illumina, Inc., San Diego, CA, USA) [40] using the following primer sequences, appended to Illumina adaptor (underlined):

Forward primer: 5’-TCGTCGGCAGCGTCAGATGTGTATAAGAGACAGCCTACGGGNGGCWGCAG 3’.

Reverse primer: 5’GTCTCGTGGGCTCGGAGATGTGTATAAGAGACAGGACTACHVGGGTATCTAATCC-3’.

PCR reactions were performed in volumes of 100 µL using OnePCRTM Ultra Master Mix (GeneDirex^®^, Taichung City, Taiwan, Cat. No. MB208-0100). PCR was performed in a thermal cycler (BIORAD T100 Thermal Cycler) with the following conditions: Initial denaturation at 95 °C for 3 min, followed by 25 cycles of denaturation at 95 °C for 30 s, annealing at 55 °C for 30 s, and extension at 72 °C for 30 s. Then, final extension at 72 °C for 5 min. Size and quality of PCR amplification products was verified with agarose gel electrophoresis. The primer pair sequences used for the V3–V4 region were employed to create a single amplicon of approximately 550 bp. Amplicons from negative controls were analyzed to detect false-positive reactions due to contaminants in swabs, extraction reagents, and PCR mixture. Library preparation and high-throughput Illumina MiSeq paired-end 2 × 300 base sequencing was performed at IGA Technology Services (Udine, Italy). Negative controls, including an empty swab for DNA extraction and amplified nuclease-free water without extracted DNA, were also sequenced and analyzed along with the samples.

### 2.7. Bioinformatic Analysis

Raw sequences were imported into Quantitative Insights into Microbial Ecology 2 (QIIME2 version 2021.4) [33] as Casava 1.8 paired-end demultiplexed fastq format. Analysis began with generating interactive positional quality plots (Appendix A) by randomly subsampling 10,000 out of 5,930,186 sequences without replacement. This was done using the QIIME2 command “qiime demux summarize”. Positional quality plots were then used to decide the best positions for trimming and truncation. Then, we used DADA2 [41] through qiime dada2 plugin for filtering out noisy sequences, correcting errors in marginal sequences, removing chimeric sequences, removing singletons, joining denoised paired-end reads, and then dereplicating these sequences producing unique amplicon sequence variants (ASVs) feature table. The following parameters were used for DADA2 pipeline: forward read sequences were truncated at 300, while reverse read sequences were truncated at 260. The first 24 bases at 5′ end of each sequence were trimmed for both forward and reverse sequences [42]. Taxonomy assignment of ASVs was performed based on trained RDP’s naive Bayesian classifier at 99% sequence similarity against the Greengenes [43] database (https://data.QIIME2.org/2021.4/common/gg-13-8-99-nb-classifier.qza, accessed on 11 August 2022). Further taxonomic classification of unrecognized features in our data set was performed using NCBI-BLAST links, which were complementary to QIIME2 and provided with dada2-rep-seq.qzv visualization file produced from DADA2 pipeline. Sequences were rarefied with QIIME2 script “qiime diversity alpha-rarefaction’’ command. Alpha diversity (within sample variation) and beta diversity (between sample variation) analysis were performed through running the QIIME2 scripts “qiime diversity core-metrics-phylogenetic” command. Statistical significance and visualization of violin plots of the alpha diversity indices (Observed, Chao1, Shannon, and Simpson) and Pielou’s evenness was performed using “ggstatsplot” R package, using the function “ggbetweenstats”; the *p* value was measured by pairwise Kruskal–Wallis test and adjusted using Holm adjustment. Dunn pairwise test was also performed. Beta diversity distances, including Bray–Curtis, Jaccard, and unweighted and weighted UniFrac distances, were represented via generating principal coordinates analysis (PCoA) plots at feature level through MicrobiomeAnalyst [44,45] using analysis of group similarities (ANOSIM) statistical method. Pairwise permutational multivariate analysis of variance (PERMANOVA) test in QIIME2 using pseudo-F statistic was also done. This was applied to measuring the amount of variation in beta diversity between groups. A *p* value < 0.05 is considered statistically significant. Taxonomy bar plots, core microbiome, and discriminative taxa between study groups were analyzed using MicrobiomeAnalyst (https://www.microbiomeanalyst.ca/, accessed on 15 December 2022) [44,45] and “ggplot2” R package of R (v 4.2.0). Discriminative taxa between study groups were measured by linear discriminant analysis (LDA) effective size (LEfSe). LDA score ≥ 2 and a *p* value < 0.05 was considered statistically significant. Correlations between taxa across pregnancy trimesters and non-pregnancy state were analyzed by Spearman correlation analysis using the “rcorr” function of R package “Hamsic” (r ≥ ±0.6, *p* ≤ 0.05). The “ggpairs” function of the R package “GGally” was used for plotting correlation between pairs of the vaginal microbiota during each trimester of pregnancy at phylum, genus, and species levels. 

### 2.8. Community State Types (CST) Assignment

Taxonomic profiles of our vaginal bacterial communities were classified into CSTs using VALENCIA (VAginaL community state typE Nearest CentroId classifier) [46]. This is a nearest centroid-based tool, which sorts samples according to their similarity to a set of reference centroids. 

### 2.9. Data Availability

The raw sequences were submitted in the National Center for Biotechnology Information (NCBI) Sequence Read Archive (SRA), under the BioProject ID PRJNA877217 https://www.ncbi.nlm.nih.gov/sra/PRJNA877217 (accessed on 15 September 2022) involving accession numbers from SAMN30696179 to SAMN30696228.

## 3. Results

### 3.1. Sequencing Data Profiles

A total of 4,051,265 read counts were maintained after the quality trimming of Illumina MiSeq raw sequencing data of 48 vaginal samples. The average read counts per sample was 84,401, while minimum and maximum counts per sample were 36,027 and 149,731, respectively. The total number of ASVs features that remained was 927. The plateau was reached at 20,000 within the rarefaction curves (Appendix A), confirming the appropriate sampling depth used. The p-max-depth was 41,064. Stacked bar charts showing taxa relative abundance for individual samples are shown in Appendix A.

### 3.2. Taxonomical Classification, Core Microbiome and Bacterial Biomarkers

#### 3.2.1. Taxonomical Classification

*Firmicutes, Actinobacteria*, and *Tenericutes* were the most abundant phyla in all study groups (Figure 1A, Appendix A). The relative abundance of *Firmicutes* was the highest in the first trimester (97.7%), compared to the second trimester (96.2%), and non-pregnant control group (92.4%), while it was the least abundant in the third trimester (86.3%). However, the relative abundance of *Actinobacteria* and *Tenericutes* were the highest in the third trimester (6.4%, 4.4%, respectively) compared to the second trimester (1.4%, 0.9%, respectively) and non-pregnant control group (5.2%, 0.9%, respectively), while they were relatively the least abundant in the first trimester (1.2%, 0.4%, respectively). 

The most abundant genera (Figure 1B, Appendix A) were *Lactobacillus*, *Gardnerella, Ureaplasma*, and *Streptococcus*. The relative abundance of *Lactobacillus* was the highest in the first trimester (95.4%) compared to the second trimester (94.7%) and non-pregnant control group (89.2%), while it was the least abundant in the in third trimester (81.8%). However, the relative abundance of *Gardnerella* and *Ureaplasma* was the highest in the third trimester (5.9%, 4.4%, respectively) when compared to the second trimester (1.3%, 0.9%, respectively) and non-pregnant control group (2.6%, 0.9%, respectively), while their relative abundance was the least in the first trimester (0.9%, 0.4%, respectively). The relative abundance of the genus *Streptococcus* was the highest in the third trimester (2.1%) when compared to the first trimester (0.5%) and non-pregnant control group (0.6%), while it was the least in the second trimester (0.05%). 

On the species level (Figure 1C, Appendix A), *L. iners* was the most abundant species in all study groups. However, it was relatively the most abundant species in the second trimester (65.3%) and non-pregnant control group (66.6%) when compared to the first trimester (52.2%), while it was relatively the least abundant in the third trimester (33%). Other detected *Lactobacillus* species included *L. helveticus, L. jensenii, L. gasseri, L. coleohominis*, and *L. reuteri*. The relative abundance of *L. helveticus* was the highest in the third trimester (25.4%) when compared to the second trimester (18.8%) and first trimester (8.4%), while its relative abundance was the lowest in the non-pregnant control group (2.9%). The relative abundance of *L. jensenii* was the highest in the first trimester (23.1%) when compared to the third trimester (16.1%) and non-pregnant control group (13.4%), while its relative abundance was the least in the second trimester (9.9%). The relative abundance of *L. gasseri* was the highest in the first trimester (11.3%) when compared to the third trimester (6.3%) and non-pregnant control group (5.9%), while its abundance was the least in second trimester (0.17%). *L. coleohominis* and *L. reuteri* were detected as minor *Lactobacillus* species. The relative abundance of *L. coleohominis* ranged from 0.22% in the third trimester compared to 0.19% in the first trimester, while its relative abundance was the least in the second trimester and non-pregnant control group (0.09%, 0.085%, respectively). The relative abundance of *L. reuteri* was 0.49% the in third trimester, 0.42% in the second trimester, and 0.29% in the non-pregnant control group, while its abundance was the least in the first trimester (0.2%).

Among the non-*Lactobacillus* species (Figure 1D, Appendix A), *Bifidobacterium breve, Peptostreptococcus anaerobius, Streptococcus anginosus, Ureaplasma parvum,* and *Streptococcus agalactiae* were frequently detected. The relative abundance of *B. breve, P. anaerobius*, and *S. anginosus* was the highest in the non-pregnant control group. The relative abundance of *B. breve* was 2.5% in the non-pregnant control group compared to 0.04% in the first trimester, while it was nearly absent in the second and third trimesters. Similarly, the relative abundance of *P. anaerobius* was 0.29% in the non-pregnant control group compared to 0.28% in the first trimester, while its relative abundance was the lowest in the second and third trimesters (0.02%, 0.019%, respectively). The relative abundance of *S. anginosus* was 0.48% in the non-pregnant control group compared to 0.12% in the third trimester, while its relative abundance was as its lowest in the second and first trimesters (0.03%, 0.01%, respectively). However, the relative abundance of *U. parvum* and *S. agalactiae* was the highest in the third trimester. The relative abundance of *U. parvum* was 4.4% in the third trimester compared to 0.9% in the second trimester and the non-pregnant control group, while its relative abundance was the lowest in the first trimester (0.4%). The relative abundance of *S. agalactiae* was 1.9% in the third trimester compared to 0.5% in the first trimester, while its relative abundance was the lowest in the non-pregnant control group (0.005%) and it was absent in the second trimester.

According to Ravel et al., 2011 [15], CST classifications using VALENCIA [46] are shown in Figure 2 panels A and B, Table 1 and Appendix A. The *L. iners*-dominant CST III was most commonly assigned in all study groups, representing more than two-thirds of our samples (32/48, 66%). This was followed by CST V (7/48, 14.58%), CST IV (5/48, 10.4%), and CST II (4/48, 8.3%). CST III represented 91.67% of the samples in the second trimester (11/12) compared to 75% (9/12) in the non-pregnant control group, and half of the samples in both the first trimester (50%, 6/12) and third trimester (50%, 6/12). Assignments to CST III-A (*L. iners* highly dominant) were observed in 31.25% (15/48) of our samples, including 16.67% (2/12) in the third trimester, 41.67% (5/12) in the second trimester, 33.33% (4/12) in the first trimester compared to 33.33% (4/12) in the non-pregnant control group. Meanwhile, assignments to CST III-B (slightly lower *L. iners* abundance) were detected in 35.42% (17/48) samples, involving 33.33% (4/12) in the third trimester, 50% (6/12) in the second trimester, and 16.67% (2/12) in the first trimester compared to 41.67% (5/12) in the non-pregnant control group. CST assignments to CST II, *L. gasseri*-dominant, were observed in 16.67% (2/12) in the third trimester, 8.33% (1/12) in the first trimester and the non-pregnant control group, and it was not recorded in the second trimester. Assignments to CST V, *L. jensenii*-dominant, were detected in 25% (3/12) in the third trimester, 16.67% (2/12) in the first trimester, and 8.33% (1/12) in the second trimester and the non-pregnant control group. For non-*Lactobacillus*-dominated CST, assignments to CST IV_C (diverse facultative and strictly anaerobic bacteria dominance) were detected in 16.67% (2/12) of the first trimester, 8.33% (1/12) in the third trimester and non-pregnant control group, and was not found in the second trimester. CST IV-C0 (slightly abundance of *Prevotella*) was detected in 8.33%(1/12) of the first trimester, CST IV-C1 (*Streptococcus* dominated) was observed in 8.33% (1/12) of the third trimester, and CST IV-C3 (*Bifidobacterium* dominated) was detected in 8.33% (1/12) of first trimester and 8.33% (1/12) of the non-pregnant control group, while CST IV_B (high relative abundance of *G. vaginalis*) was only detected in 8.33% (1/12) in the first trimester.

#### 3.2.2. Bacterial Biomarkers

Bacterial biomarkers of each study group were detected through Linear discriminant analysis effect size (LEfSe) using MicrobiomeAnalyst. At species level, *B. breve* was detected to be significantly abundant in the non-pregnant control group compared to the pregnant group. However, across pregnancy trimesters, *L. gasseri* was significantly abundant in first trimester and *S. agalactiae* in third trimester. 

### 3.3. Alpha Diversity

Alpha diversity, as calculated by the Alpha diversity indices Observed (*p* = 0.62, Figure 3A), Chao1 (*p* = 0.62, Figure 3B), Shannon (*p* = 0.19, Figure 3C), and Simpson (*p* = 0.18, Figure 3D), revealed non-significant differences in the vaginal community composition within study groups. This was further confirmed by Pielou’s evenness alpha diversity (*p* = 0.22, Figure 4, Appendix A). There was a relative homogeneity of the vaginal microbiome both across the three trimesters of pregnancy and, in comparison, to the non-pregnant status. 

### 3.4. Beta Diversity

The dissimilarities between vaginal microbial communities were detected through measuring beta diversity by calculating the Bray–Curtis and Jaccard indices. Significant differences in the composition of the vaginal microbial communities among study groups were revealed by Bray–Curtis (R = 0.059326, *p* < 0.037, ANOSIM) and Jaccard (R = 0.059326, *p* < 0.037, ANOSIM) dissimilarity distances (Figure 5 panels A and B). These differences were further confirmed by pairwise PERMANOVA tests. There was a significant difference in the composition of the vaginal microbial communities between all study groups except between the vaginal microbial communities of the second and third trimester of pregnancy as they were revealed to be relatively similar when compared to the first trimester and the non-pregnant state. However, phylogenetic dissimilarity distances (Figure 5 panels C and D), unweighted UniFrac (R = 0.014871, *p* < 0.3, ANOSIM), and weighted UniFrac tests (R = 0.01908, *p* < 0.151, ANOSIM) revealed non-significant differences between study groups confirming the low diversity of the vaginal microbiome during pregnancy.

### 3.5. Correlation of Vaginal Microbiota during Pregnancy

Correlation analysis revealed that *Firmicutes* was negatively correlated with almost all other phyla (Figure 6A, Appendix A). Likewise, *Lactobacilli* were negatively correlated with almost all other genera (Figure 6B, Appendix A) showing significant negative correlation with the genera *Dialister* and *Prevotella*. Similarly, *L. iners* was negatively correlated with almost all other species (Figure 6C, Appendix A). There was a significant negative correlation between *L. iners* and other *Lactobacillus* species, in particular, *L. helveticus*. There was also a negative correlation between *L. iners* and other non-*Lactobacillus* species, demonstrating significant negative correlations with *G. vaginalis, S. agalactiae, P. anaerobius*, and *S. anginosus* versus a significant positive correlation between the non-lactobacillus species pairs *G. vaginalis* and *S. agalactiae, G. vaginalis* and *P. anaerobius, S. aginosus* and *P. anaerobius.* Further positive correlations were found between *Finegoldia magna* and the species *S. agalactiae, P. anaerobius*, and *S. aginosus*.

## 4. Discussion

A deep investigation of the vaginal environment can provide new perspectives for future health through understanding the pathophysiology of pregnancy and identifying women at risk for adverse pregnancy outcomes. In the current study, we have characterized the vaginal microbiome during normal pregnancy at the three gestational trimesters compared to non-pregnant status, in Ismailia, Egypt through profiling the vaginal microbiota using 16S rRNA next-generation sequencing. We were able to describe fluctuations in the composition of the vaginal microbiota during the three trimesters of pregnancy compared to non-pregnant state.

In general, irrespective to the period, pregnancy was dominated by members of the phylum *Firmicutes*. As expected, *Lactobacillus*-dominant profiles were a characteristic feature of healthy vaginal microbiomes during pregnancy. The relative abundance of the genus *Lactobacillus* was in a range from 81% to 95%, confirming the low biodiversity of healthy vaginal bacterial communities. This is in agreement with other populations worldwide [6,9,10,12,17,18,19,28,30,47] despite differences in methodologies and definitions. The growth-inhibiting ability of vaginal lactobacilli is well-documented and primarily attributed to their role in lowering the vaginal pH by lactic acid production [3,24,48]. Other reported growth-inhibiting strategies included hydrogen peroxide production and release of antimicrobial substances, such as bacteriocins [3,24,48,49,50].

*L. iners* was the most abundant species in both healthy pregnant and non-pregnant women in our study. *L. iners* has been frequently reported as the most *Lactobacillus*-dominant member of the vaginal microbiome of women with African ancestry. A PRISMA-compliant review conducted by Juliana and co-workers [5] in seven sub-Saharan African countries identified *L. iners* among the most prevalent species. Similarly, Gautam et al., 2015 [51] displayed that *L. iners*-dominated vaginal microbiome clusters among the most dominant clusters in African women from four sub-Saharan countries. Another study by Jespers et al., 2015 [52] in African women from three sub-Saharan regions demonstrated the higher relative abundance of *L. iners* in women with normal Nugent score compared to non-African studies. Likewise, Serrano et al., 2019 [10] showed pregnant women of African ancestry with significantly higher representation of *L. iners* compared to non-African ancestry. 

Another species that was commonly detected as a characteristic feature of African vaginal microbiomes is Ca. *Lachnocurva vaginae* or BVAB1 [15,27]. This bacterium is consistently correlated with pregnancy complications, such as preterm birth [10,35]. However, Ca. *Lachnocurva vaginae* was not detected among our vaginal microbial communities even when compared to studies using similar approaches for investigating the vaginal microbiome profiles [53,54]. 

In consistence with others [8,15,38], our correlation analysis demonstrated that the phylum *Firmicutes*, the genus *Lactobacillus*, and the species *L. iners* were negatively correlated with almost all other vaginal taxa. According to Ng et al., 2021 [8], this emphasizes their well-documented exclusionary behavior in the vaginal ecosystem. Furthermore, we found a relative increase in the abundance of *Gardnerella* and *Ureaplasma* in the third trimester compared to the first and second trimesters, but this increase was in accordance with a high *Lactobacillus* abundance. *Gardnerella* and *Ureaplasma* are well-known pathogens associated with adverse pregnancy outcomes [55]. However, recent microbiome reports [7,55] point out that vaginal colonization with *Ureaplasma* and *Gardnerella* during a pregnancy with *Lactobacillus* abundance seems to be protective against pregnancy complications. Park et al., 2022 [55] demonstrated that *Ureaplasma* and *Lactobacillus* coexistence with high *Lactobacillus* abundance leads to term birth. In addition, Park et al., 2022 [55], Liu et al., 2022 [7], and DiGuilio, 2015 [16] reported that preterm birth and premature membrane ruptures are only predictive with low or depleted *Lactobacillus* abundance in the presence of *Ureaplasma* and *Gardnerella.* This suggests that the balanced co-colonization of *Ureaplasma* and *Gardnerella* in the vaginal ecosystem under *Lactobacillus* dominance may promote full-term birth and prevent pregnancy complications.

We also found *L. iners* was negatively correlated with *L. helveticus*, which was common among our vaginal samples. This could be attributable to the overlapping ecological functions in the vaginal econiche. On the other hand, we detected multiple positive correlations between bacteria associated with bacterial vaginosis or vaginal dysbiosis. We found positive correlations between the pairs of *G. vaginalis* and *S. agalactiae, G. vaginalis* and *P. anaerobius*, and *S. aginosus and P. anaerobius*. Further positive correlations were found between *F. magna* and *S. aginosus*, as well as *S. agalactiae* and *P. anaerobius*. This is in line with previous reports which highlighted the importance of the synergistic interplay between non-*lactobacillus* vaginal taxa leading to *Lactobacillus* displacement, overgrowth of anaerobic bacteria, rise in vaginal pH, and ultimately vaginal dysbiosis or bacterial vaginosis [56,57,58,59,60,61].

Microbiota profiling through 16S rRNA gene next-generation sequencing is primarily dependent on selecting suitable primer pairs and hypervariable regions of the bacterial 16S rRNA gene [62]. Primer selection may have significant influence on obtaining different results [62]. In the current study, we used universal primers, as per instructions of the standard Illumina protocol [40] spanning the V3–V4 hypervariable region, which is widely used for investigating the human vaginal microbiome [1,2,4,7,12,13,38]. This region has been reported to provide intense discrimination between most microbiota of the urogenital tract [7,13]. However, alternative hypervariable regions of the 16S rRNA were also utilized, such as the V3–V5 [16,28,30] and the V1–V2 or V1–V3 hypervariable regions [62,63]. Prince et al., 2015 [63], indicated primers spanning the V1–V3 region are more suitable for detecting *Lactobacillus, Prevotella*, and *Clostridium* species, whereas primers spanning the V3–V5 region are better when considering *Enterobacteriaceae* and *Bifidobacteriaceae*. According to Graspeuntner et al., 2018 [62], a greater number of taxa in the vaginal microbiota are detectable under the V3–V4 region compared to V1–V2, which is inappropriate for identifying *G. vaginalis*. The authors indicated that when considering well-defined vaginal microbiota with proven influence in both health and disease, such as *G. vaginalis, Bifidobacterium bifidum*, and *Chlamydia trachomatis*, the V3–V4 region is recommended for 16S rRNA microbiome studies. These taxa are underestimated in V1–V2 16S rRNA vaginal community-based profiles, resulting in failure to correctly assess the bacterial diversity and overestimation of the abundance of other taxa. 

The selection of tools and databases for appropriate taxonomic assignation is another important methodological issue that should be considered in 16S rRNA microbiome studies. Similar to others [1,8,38], we used QIIME 2 software and Greengenes database. QIIME 2 has been demonstrated to facilitate the reproducible and comprehensive analysis of diverse microbiome data [64]. Greengenes is a popular taxonomic database, which is widely used in profiling the human vaginal microbiome [65,66,67,68,69,70]. Nonetheless, microbiota profiling by 16S rRNA next-generation sequencing is the preferred approach when compared to traditional culture. Currently, it is the quickest and most accurate means for investigating the human vaginal microbiota.

The present study has several limitations that should be avoided in the future. The small sample size of enrolled participants may not give an accurate demonstration of the composition and diversity of the vaginal microbiomes. In addition, we were unable to collect detailed clinical data that would have been valuable in determining potential correlations based on pregnancy state and associated vaginal community composition. Moreover, 16S rRNA next-generation sequencing was used for profiling vaginal bacterial communities as it allows the simultaneous detection of many taxa. However, this technique has limited taxonomic power, particularly on the species level and is not precise enough when considering taxa with a minor abundance [71,72,73]. Some authors may use species-specific quantitative real-time PCR assays to confirm the identity and relative abundance [74]. However, this approach is hampered by the limited multiplex capability of real-time PCR [75,76]. Alternative approaches, such as whole genome shotgun metagenomic sequencing (WGS), should be considered in future studies. WGS is more precise in revealing bacterial species, diversity, and abundance with added the advantage of providing direct information on functional genes [71,72,73]. 

## 5. Conclusions

In conclusion, we investigated the vaginal microbiome composition of healthy pregnant women in Ismailia, Egypt during the first, second, and third trimesters compared to healthy non-pregnant women. As expected, phylum *Firmicutes* and genus *Lactobacillus* dominated during the three trimesters of pregnancy. *L. iners*-dominant vaginal communities CST-III prevailed as *L. iners* was the most abundant species in both healthy pregnant and non-pregnant women in our study. *L. iners* was almost negatively correlated with all other species in the vagina, in particular, with *G. vaginalis, S. anginosus*, and *S. agalactiae*. In contrast, we found significant positive correlations between the species pairs *G. vaginalis* and *S. agalactiae, P. anaerobius* and *F. magna, S. aginosus* and *F. magna, S. agalactiae* and *P. anaerobius.* This highlights the protective role of lactobacilli in promoting vaginal health and emphasizing the potential synergistic interplay among vaginal microbiota to induce vaginal dysbiosis. However, a major limitation of the current study is the small sample size and minimal demographic data. Further studies recruiting larger cohorts with detailed demographics of participants should be considered.

## Figures and Tables

**Figure 1 microorganisms-11-00139-f001:**
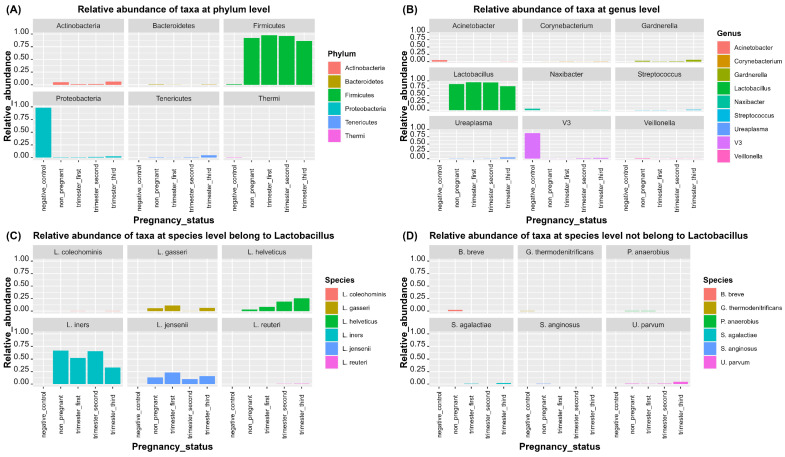
Relative abundance of taxa across the three trimesters of pregnancy in comparison to non-pregnant state. (**A**) Relative abundance of taxa at phylum level. (**B**) Relative abundance of taxa at genus level. (**C**) Relative abundance of *Lactobacillus* species. (**D**) Relative abundance of non-*Lactobacillus* species (Figure 1D is re-represented as Appendix A with different y-scale to show the relative abundance of minor non-*Lactobacillus* species). Negative control represents samples with empty swab for DNA extraction and amplified nuclease-free water without extracted DNA.

**Figure 2 microorganisms-11-00139-f002:**
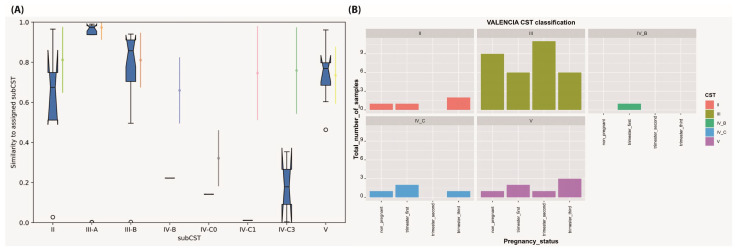
Community State Assignment (CST) of vaginal microbiomes across the three trimesters of pregnancy in comparison to non-pregnant state using VALENCIA. (**A**) Similarity of vaginal samples to the assigned CST and subCST designations. (**B**) CST classification of the vaginal samples, as revealed by VALENCIA.

**Figure 3 microorganisms-11-00139-f003:**
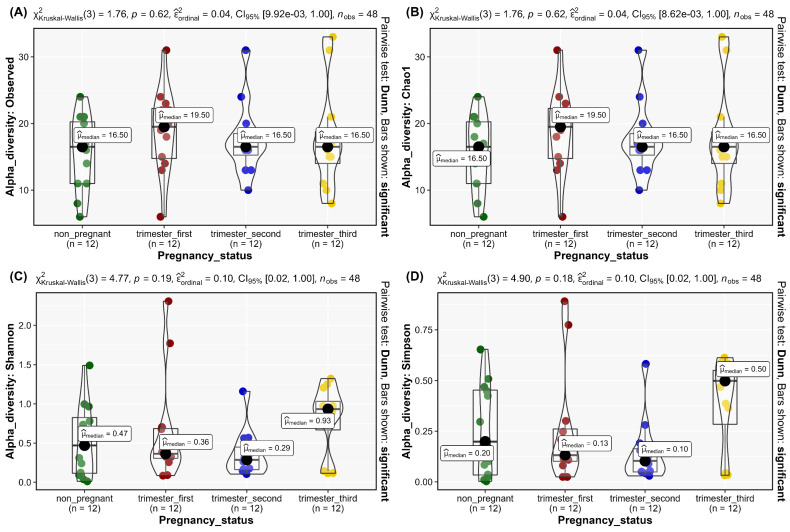
Alpha diversity indices of vaginal microbial communities across the three trimesters of pregnancy in comparison to non-pregnant state. Alpha diversity, as calculated by the (**A**) Observed (**B**) Chao1 (**C**) Shannon (**D**) Simpson Alpha diversity indices. A *p* value < 0.05 is considered statistically significant, as calculated by pairwise Kruskal–Wallis test. Dunn pairwise test was performed.

**Figure 4 microorganisms-11-00139-f004:**
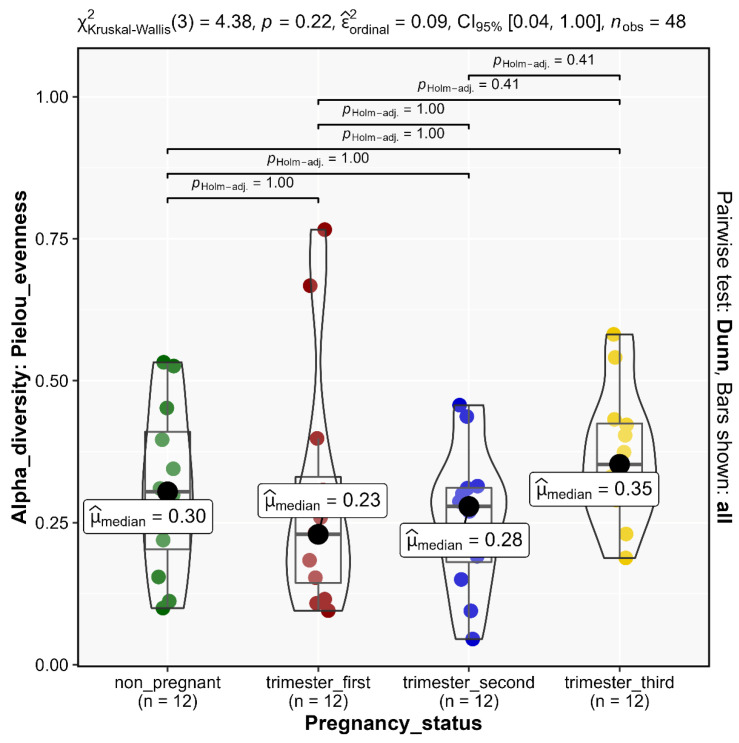
Pielou’s evenness Alpha diversity of vaginal microbial communities across the three trimesters of pregnancy in comparison to non-pregnant state. Holm-adjusted *p* values were used to indicate significance. Dunn pairwise test was performed.

**Figure 5 microorganisms-11-00139-f005:**
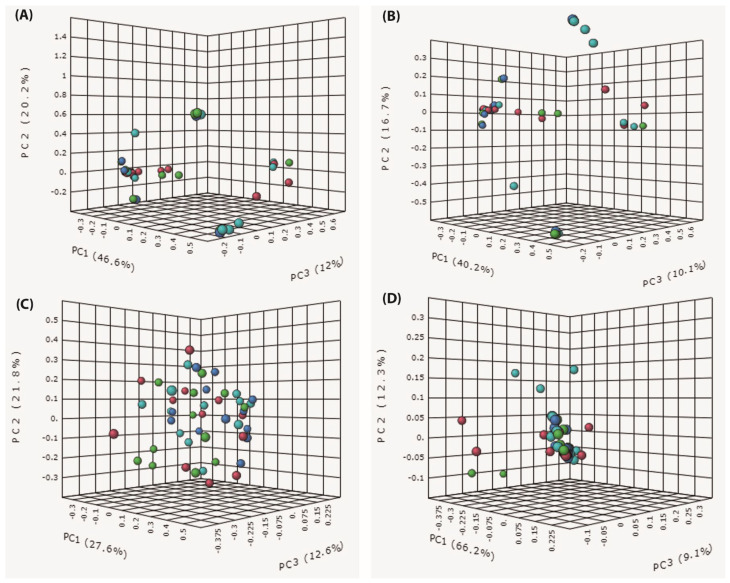
Beta diversity indices shown as two-dimensional principal coordinate analysis (PCoA) plots of vaginal microbial communities across the three trimesters of pregnancy in comparison to non-pregnant state. Beta diversity was calculated by (**A**) Bray–Curtis index and (**B**) Jaccard index (**C**) unweighted UniFrac test (**D**) weighted UniFrac test. A *p* value < 0.05 is considered statistically significant.

**Figure 6 microorganisms-11-00139-f006:**
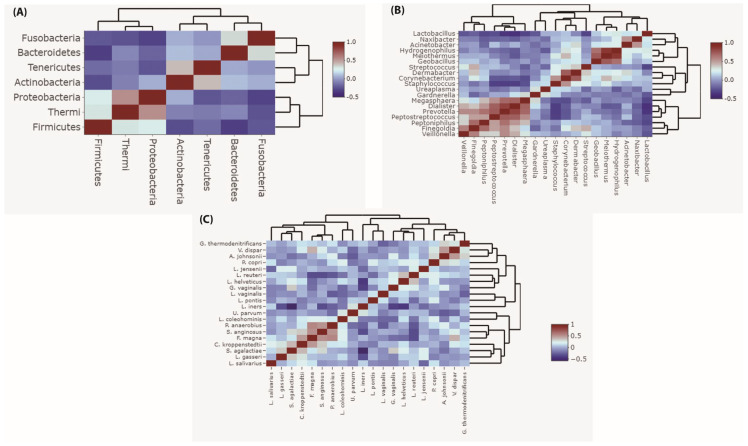
Correlation plot for dominant taxa during pregnancy. (**A**) Correlation plot for dominant phyla. (**B**) Correlation plot for dominant genera. (**C**) Correlation plot for dominant species. Positive correlation strength relationship is shown in red and negative correlation strength relationship is shown in blue. The plot was generated using Spearman’s correlation coefficients between each taxon and all other taxa.

**Table 1 microorganisms-11-00139-t001:** Distribution of samples in each community state-type according to pregnancy status.

CST/Pregnancy Status	II	III		IV-B	IV-C			**V**	**Total**
		III-A	III-B		IV-C0	IV-C1	IV-C3		
Non-pregnant control group	1 (8.33%)	4 (33.33%)	5 (41.67%)	-	-	-	1 (8.33%)	1 (8.33%)	12
Trimester—first	1 (8.33%)	4 (33.33%)	2 (16.67%)	1 (8.33%)	1 (8.33%)	-	1 (8.33%)	2 (16.67%)	12
Trimester—second	-	5 (41.67%)	6 (50%)	-	-	-	-	1 (8.33%)	12
Trimester—third	2 (16.67%)	2 (16.67%)	4 (33.33%)	-	-	1 (8.33%)	-	3 (25%)	12
Total	4 (8.33%)	15 (31.25%)	17 (35.42%)	1 (2.08%)	1 (2.08%)	1 (2.08%)	2 (4.17%)	7 (14.58%)	48

## Data Availability

The data presented in this study are openly available in the National Center for Biotechnology Information (NCBI) Sequence Read Archive (SRA), under the BioProject ID PRJNA877217 https://www.ncbi.nlm.nih.gov/sra/PRJNA877217 (accessed on 15 September 2022) involving accession numbers from SAMN30696179 to SAMN30696228.

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
