# Peer review of "Compositional Changes in the Vaginal Bacterial Microbiome of Healthy Pregnant Women across the Three Gestational Trimesters in Ismailia, Egypt"

_microorganisms, 2023, doi:10.3390/microorganisms11010139_

Round 1

Reviewer 1 Report

Few questions that need to be addressed:

Were the community-based types used to classify the subjects based on pregnant or nonpregnant women (the ones used in the database)?

Were there any differences between the 12 individual samples of the control (were the control showing any signs of similarity) to attribute the variations to pregnancy and rule out any individual variations?

Is there a way to present alpha diversity indices in numbers (for each sample) to be able to indicate the extent of diversity within each sample?

Figure 1 (A through D) what is the negative control? (what sample was it from?)

Figures 1D: the figure is not showing any bars almost. Does it need a different scale? Is it correct this way?

Labels on the axes of figure 2B are not clear.

Figures 5 and 6 need to be enlarged and presented at a better resolution as the data cannot be read. I strongly recommend changing the format of figure 5 to better represent the relationship between the samples and each other or other samples (in a 3D presentation not as such).

Figures 7 and 8 should be removed as they are totally not clear and do not allow proper comparisons between Phyla/ genera/ Lactobacilli or non-Lactobacilli. Simply explain the results in the text.

Author Response

Detailed point by point response to Reviewer-1

First of all, we would like to thank the Reviewer for his effort in reviewing the manuscript and providing the valuable comments.

-Were the community-based types used to classify the subjects based on pregnant or nonpregnant women (the ones used in the database)?

Response: All the women included in the database of VALENCIA were non-pregnant. However, Jacques Ravel working group are the authors of VALENCIA. Ravel`s et al 2011 (10.1073/pnas.1002611107) was the first to categorize vaginal bacterial communities into CST types. Regardless of pregnancy state, VALENCIA is applicable for the classification of both pregnant and non-pregnant vaginal communities. Example of studies utilized VALENCIA for pregnant vaginal microbiomes:

  • DOI: 10.1038/s41598-022-20608-7
  • DOI: 10.1128/msystems.00017-22

-Were there any differences between the 12 individual samples of the control (were the control showing any signs of similarity) to attribute the variations to pregnancy and rule out any individual variations?

Response: We tried as much as possible to match all participants using the inclusion and exclusion criteria so that pregnancy state is the only variable which we used to subdivide samples into four groups: non-pregnant, Pregnant-1st trimester, pregnant 2nd trimester, pregnant third trimester. We also checked that age was matched across the four groups with one-way ANOVA and between pregnant and non-pregnant participants with one-tailed t-test. Both were non-significant between groups (A Supplementary containing age of participants with p values for ANOVA and t-test was provided as Table S1). The vaginal pH was measured for the non-pregnant control subjects but not for the pregnant subjects as recommended by Agaard et al 2012 (doi: 10.1371/journal.pone.0036466.).

-Is there a way to present alpha diversity indices in numbers (for each sample) to be able to indicate the extent of diversity within each sample?

Response: Alpha diversity indices were automatically generated as a single value for each group as indicated by violin plots then differences between groups are evaluated for statistical significance. In order to ease comparisons, we added a stacked bar chart showing taxa relative abundance of individual samples where each bar represents one sample. Kindly check supplementary figure S3.

-Figure 1 (A through D) what is the negative control? (what sample was it from?)

Response: The included negative control is for samples with an empty swab used in parallel with actual samples during DNA extraction, PCR and next-generation sequencing. The figure legend has been modified to make this clear.

-Figures 1D: the figure is not showing any bars almost. Does it need a different scale? Is it correct this way?

Response: All non-Lactobacillus species had relative abundances less than 0.05 in comparison to other Lactobacillus species. The scale used was the best to express the relative abundance for all taxa on the levels: Phyla, Genus, Lactobacillus species and non-lactobacillus species. We did not use a different scale for figure 1D so that there is no false positive impression with a large relative abundance of non-lactobacillus species and also to be consistent with the rest of the figures panels for easier comparison and result interpretation. However, we have added a representation of Figure 1D with a different scale to show the relative abundance of minor non-Lactobacillus species. Kindly check supplementary figure S4.

-Labels on the axes of figure 2B are not clear.

Response: Done as recommended. Labels have been modified.

-Figures 5 and 6 need to be enlarged and presented at a better resolution as the data cannot be read. I strongly recommend changing the format of figure 5 to better represent the relationship between the samples and each other or other samples (in a 3D presentation not as such).

Response: Done as recommended. Figures have been modified.

-Figures 7 and 8 should be removed as they are totally not clear and do not allow proper comparisons between Phyla/ genera/ Lactobacilli or non-Lactobacilli. Simply explain the results in the text.

Response: Done as recommended. Figures have been removed.

Reviewer 2 Report

General comment

l  As the author mention, the state of the vaginal microbiome is important to women’s health and pregnancy. Also, it can be different by culture, region, and behavior, so this study can contribute to women’s health in Egypt. Furthermore, the description of vaginal microbiome results with the location in the discussion part is attractive.

l  To improve the manuscript's quality more informative, I’m going to suggest my opinion to the author of this manuscript. Honestly, reviewer don't know the actual author's research environment and circumstance. Furthermore, sometimes, there may be limitations in sample collection and securing clinical data in these kinds of studies. But I hope you reflect my comment as much as possible.

-      It needs to change the size and resolution of text and figures in the whole manuscript.

-      Does the author have clinical data and survey results on participant enrollment? If it is, please add or mention these to the corresponding part in the manuscript, even as supplementary data.

-      Even if I’m not a medical doctor, I can understand the clinical procedure for classifying all participants into four groups. But, if the author can, please describe the clinical procedure for classifying participants based on the medical reference. (Even though there are a lot of researchers in a similar research field, it may need for the purpose that this method is proper for classifying patients. If it is correct, this will give your method credibility.)

-      Because the results of vaginal microbiome studies targeting Egyptian women are rare, this study is considered meaningful. To become a more informative study, additional results related to the correlation of metadata (behavior, eating habit, or estrogen level) and vaginal microbiome may need.

Specific comment

1)    introduction

-      I think this part is well-demonstrated for final goal of this manuscript.

2)    Materials & Methods

L145~146 : What criteria is used for subsampling 10,000 out of 5,930,186 sequences? Does a  value “10,000” was selected based on the rarefaction curve? If it is, please describe this in the corresponding sentence.

L154~156 : I think there is no significant difference in data even if the author uses the Greengene database with 99% sequence similarity. But does any reason that you use 97% Greengene database? Greengene database with 99% sequence similarity also can be secure easily.

L160~169 : The statistical analysis method was distinguished based on the normality and variance of the dataset. Please mention normality and variance tests for statistical analysis.

3)    Results

-      In CST, III-A, III-B, IV-C1, IV-C1, and IV-C3 were divided according to an abundance of representative genus or species corresponding to specific groups. What is a criterion for abundance value to divide CST type?

-      If there is estrogen concentration in all participants, does any correlation between estrogen concentration and specific bacteria or pregnancy stage? If it is, further analysis may need, and it can improve manuscript quality.

-      The comment for negative control in figure 1 should be needed. Also, please put the genus and species names together in figure 1-D.

-      Does any bacterial vaginosis in participants classified into CSV-IV and V groups during the experiment?

4)    Discussion

-      In this study, as the pregnancy stage increases, the abundance of Lactobacillus and Gardnerella & Ureaplasma are decrease and increase, respectively. Also, participant in all groups predominate in L. iners correspond to CST-III. I think the manuscript quality will be better if there is additional descriptions about these result based on the metadata or knowledge of medical field.

-      Participant enrollment proceeded from July to October 2020. If there is health information about the children(baby) of the participant, can the author describe any relationship between the baby’s health and the mother’s virginal microbiome, even if the vaginal health state of all participants is good?

L461~466

1) please put "." to "L iners"

2) I can understand there is a light negative correlation between L. iners and L. helveticus. But, the positive description of L. helveticus may confuse the general understanding of this manuscript. Please re-write this paragraph.

Author Response

Detailed point by point response to Reviewer-2

First of all, we would like to thank the Reviewer for his effort in reviewing the manuscript and providing the valuable comments.

General comment

As the author mention, the state of the vaginal microbiome is important to women’s health and pregnancy. Also, it can be different by culture, region, and behavior, so this study can contribute to women’s health in Egypt. Furthermore, the description of vaginal microbiome results with the location in the discussion part is attractive.

To improve the manuscript's quality more informative, I’m going to suggest my opinion to the author of this manuscript. Honestly, reviewer don't know the actual author's research environment and circumstance. Furthermore, sometimes, there may be limitations in sample collection and securing clinical data in these kinds of studies. But I hope you reflect my comment as much as possible.

-      It needs to change the size and resolution of text and figures in the whole manuscript.

Response: Done as recommended. The resolution of all figures has been enhanced.

-      Does the author have clinical data and survey results on participant enrollment? If it is, please add or mention these to the corresponding part in the manuscript, even as supplementary data.

Response: We only were able to collect data about the age of enrolled participants. This data has been added as a supplementary table S1. We also checked that age was matched across the four groups with one-way ANOVA and between pregnant and non-pregnant participants with one-tailed t-test. Both were non-significant between groups (p values for ANOVA and t-test were provided with Table S1). unfortunately, no other clinical data is available for the participants. We have added this to study limitations. Kindly check lines 504-508

-      Even if I’m not a medical doctor, I can understand the clinical procedure for classifying all participants into four groups. But, if the author can, please describe the clinical procedure for classifying participants based on the medical reference. (Even though there are a lot of researchers in a similar research field, it may need for the purpose that this method is proper for classifying patients. If it is correct, this will give your method credibility.)

Response: The participants were divided according to pregnancy status into four groups: Non-pregnant, pregnant during the first trimester, pregnant during the second trimester, and pregnant during the third trimester. Obstetrical dating for gestational age was determined as indicated by the American College for Obstetricians and Gynecologists (ACOG) (https://www.acog.org/). First trimester is from first day of last menstrual period (LMP) to 13 weeks and 6 days. Second trimester is from 14 weeks and 0 days to 27 weeks and 6 days. Third trimester is from 28 weeks and 0 days to 40 weeks and 6 days. We have added this information to the methods section. Kindly, check lines 94-100

-      Because the results of vaginal microbiome studies targeting Egyptian women are rare, this study is considered meaningful. To become a more informative study, additional results related to the correlation of metadata (behavior, eating habit, or estrogen level) and vaginal microbiome may need.

Response: unfortunately, no metadata is available for the participants. We have added this to study limitations. Kindly check lines 504-508

Specific comment

1)    introduction

-      I think this part is well-demonstrated for final goal of this manuscript.

2)    Materials & Methods

L145~146 : What criteria is used for subsampling 10,000 out of 5,930,186 sequences? Does a  value “10,000” was selected based on the rarefaction curve? If it is, please describe this in the corresponding sentence.

Response: According to QIIME2 manual, using the QIIME2 command “qiime demux summarize“ random subsampling is recommended to determine the positional quality scores. This command generates positional quality plots which then are used to decide the best positions for trimming and truncation in the next step. The default number for random subsampling as recommended by QIIME2 is 10000. So, this number was not based on the rarefaction curve. All our rarefaction curves plateau at 20000 meaning that all samples have been sequenced deeply enough to capture the full diversity for bacterial communities (Supplementary figure S1). The p-max-depth was 41064. We have added the positional quality plots as supplementary figures S1, lines 154-157, 212-213

L154~156 : I think there is no significant difference in data even if the author uses the Greengene database with 99% sequence similarity. But does any reason that you use 97% Greengene database? Greengene database with 99% sequence similarity also can be secure easily.

Response: We agree with the reviewer. We used 99% sequence similarity as indicated by the provided link. This was a typing error which was corrected now. Kindly check line 165.

L160~169 : The statistical analysis method was distinguished based on the normality and variance of the dataset. Please mention normality and variance tests for statistical analysis.

Response: Done as recommended. Kindly, check lines 173-183.

3)    Results

-      In CST, III-A, III-B, IV-C1, IV-C1, and IV-C3 were divided according to an abundance of representative genus or species corresponding to specific groups. What is a criterion for abundance value to divide CST type?

Response: According to the VALENCIA article, for the sub-CSTs denoted with A and B: The “A” version represents samples that had a higher relative abundance of the focal species (The Lactobacillus dominant species, more than 90%), with the “B” version representing samples with a somewhat lower relative abundance of that species (60-90%). The CST IV-A, IV-B, and IV-C do not have a high relative abundance of Lactobacilli where CST IV-A has a high relative abundance (30% or more) of Candidatus Lachnocurva vaginae (formerly known as BVAB1) and a moderate relative abundance of G. vaginalis, while IV-B has a high relative abundance of G. vaginalis (30% or more) and low relative abundance of Ca. L. vaginae. Samples assigned to CST IV-C have a low relative abundance of Lactobacillus spp., G. vaginalis, A. vaginae, and Ca. L. vaginae and were instead characterized by the abundance of a diverse array of facultative and strictly anaerobic bacteria. These are further split into 5 sub-CSTs as follows: CST IV-C0—an even community with moderate amount of Prevotella, CST IV-C1 Streptococcus dominated, CST IV-C2—Enterococcus dominated, CST IV-C3—Bifidobacterium dominated, and CST IV-C4—Staphylococcus dominated.

-      If there is estrogen concentration in all participants, does any correlation between estrogen concentration and specific bacteria or pregnancy stage? If it is, further analysis may need, and it can improve manuscript quality.

Response:  estrogen-driven maturation of the vaginal epithelium leads to the accumulation of glycogen. This glycogen deposition acts as a chemotactic agent for lactobacilli as this is a major substrate used by these microbes, which is broken to glucose and the fermented to lactic acid, thereby playing a significant role in lowering the vaginal pH.

-      The comment for negative control in figure 1 should be needed. Also, please put the genus and species names together in figure 1-D.

Response: Done as recommended.

-      Does any bacterial vaginosis in participants classified into CSV-IV and V groups during the experiment?

Response: no, all participants were healthy.

4)    Discussion

-      In this study, as the pregnancy stage increases, the abundance of Lactobacillus and Gardnerella & Ureaplasma are decrease and increase, respectively. Also, participant in all groups predominate in L. iners correspond to CST-III. I think the manuscript quality will be better if there is additional descriptions about these result based on the metadata or knowledge of medical field.

Response: Done as recommended. Kindly, check lines 447-460

-      Participant enrollment proceeded from July to October 2020. If there is health information about the children(baby) of the participant, can the author describe any relationship between the baby’s health and the mother’s virginal microbiome, even if the vaginal health state of all participants is good?

Response: Unfortunately, there is no health information about the children(baby) of the participant.

L461~466

  • please put "." to "L iners"

Response: Done as recommended.

  • I can understand there is a light negative correlation between L. iners and L. helveticus. But, the positive description of L. helveticus may confuse the general understanding of this manuscript. Please re-write this paragraph.

Response: Done as recommended. Kindly, check lines 461-466

Reviewer 3 Report

The study was an interesting read, however it was only based on one assay. Maybe in future work the bacterial communities could be validated by real time pcr to give absolute concentrations.

Author Response

Detailed point by point response to Reviewer-3

First of all, we would like to thank the Reviewer for his effort in reviewing the manuscript and providing the valuable comments.

Comments and Suggestions for Authors

The study was an interesting read, however it was only based on one assay. Maybe in future work the bacterial communities could be validated by real time pcr to give absolute concentrations.

Response: Done as recommended. We have referred to the value of real-time PCR in confirming the information obtained by 16S rRNA next-generation sequencing under study limitations. Kindly check lines 508-518.

Round 2

Reviewer 2 Report

Thank you for your effort to revise a manuscript based on the reviewer's comment. I hope you have wonderful research in the future.

In the revision, the author tried to revise their manuscript according to the reviewer's cover letter, kindly. In a revised manuscript, they mentioned several comments, especially the small number of enrolled participants and the absence of clinical data, which corresponded as a limitation of their study honestly.

Even if there is no comprehensive data analysis using metadata because of the absence of clinical metadata, this manuscript is valuable. Also, It can be a "primary study for Egyptian women's health." Furthermore, I think there is no minor error during data analysis, and all their results are supported by a bibliography in the author's manuscript.